# Ligamentotaxis Effect of Lateral Lumber Interbody Fusion and Cage Subsidence

**DOI:** 10.3390/jcm14134554

**Published:** 2025-06-26

**Authors:** Ryosuke Tomio

**Affiliations:** Department of Neurosurgery, Honjo Neurosurgery and Spine Surgery Clinic, Saitama 367-0030, Japan; tomy0807@hotmail.com; Tel.: +(81)-0495-23-9156

**Keywords:** LLIF, OLIF, ligamentotaxis, indirect decompression, cage subsidence

## Abstract

**Background/Objectives**: Lateral lumbar interbody fusion (LLIF) has gained popularity as an effective technique for indirect decompression through ligamentotaxis. Despite the perceived importance of using appropriately sized cages for achieving optimal decompression, comprehensive reports on cage size and its impact on indirect decompression are limited. This study aimed to assess the ligamentotaxis effect by measuring the “backward bulging” length in pre- and postoperative MRIs and examining its correlation with cage size and subsidence. **Methods**: T2 images of 270 patients with lumbar herniated disc and/or lumbar spondylolisthesis (June 2022 to March 2025) were analyzed for 530 intervertebral spaces. Data on gender, age, length of hospital stay, preoperative and postoperative lumbar JOA scores, and the level of the disease were collected. Measurements included backward bulging length, intervertebral height, and cage subsidence. Statistical analysis was performed using StatMate. Surgical procedures involved oblique lateral interbody fusion (OLIF) to minimize impact on the iliopsoas and lumbar plexus. Trial cages starting from 8 mm were sequentially inserted, with confirmation through lateral fluoroscopy. Posterior fixation was performed using percutaneous pedicle screws. **Results**: Analysis of 530 intervertebral spaces revealed that 70% could accommodate a cage 3 mm or larger than the preoperative intervertebral height. Significant backward bulging shortening (3 mm or more) occurred in 339 spaces, predominantly with larger cages. Only 8.8% of cases (14/159) with a large backward bulging shortening had an intervertebral height extension of 3 mm or less. On the other hand, a large reduction in backward bulging was observed in 91.3% of cases (339/371) with an intervertebral height extension of 3 mm or more. Postoperative cage subsidence was observed in 9.2% (49/530) of all intervertebral spaces and 8.6% (32/371) in spaces where a cage larger than 3 mm was used. There was no statistically significant difference between these two groups. **Conclusions**: To achieve a sufficient ligamentotaxis effect, it is necessary to select a cage size that allows for an intervertebral height increase of at least 3 mm compared to the preoperative measurement.

## 1. Introduction

Lateral lumber interbody fusion (LLIF) has become popular. LLIF is a surgical technique that offers advantages in terms of bone fusion rates and infection rates, and enables anterior fixation while preserving posterior structures such as the paraspinal muscles, spinous processes, laminae, and facet joints. In addition, indirect decompression by ligamentotaxis is effective for spinal stenosis, and it is considered particularly effective for decompressing pressure on the dural sac from the front, such as in cases of protruded hernia or lumbar spondylolisthesis [1,2,3,4,5,6,7]. If one were to point out the drawbacks of this surgical technique, they would include the risk of injury to abdominal organs, such as the intestines or to the aorta, when performed by surgeons not well versed in the procedure. Additionally, since anterior fixation of the vertebral body can be achieved without accessing the spinal canal from the posterior side, a separate posterior skin incision is required if direct decompression becomes necessary. Furthermore, because anterior fixation is typically performed in the lateral decubitus position, intraoperative repositioning from lateral to prone is necessary to perform percutaneous pedicle screw placement. At our institution, over the past three years, more than 300 cases of this surgical technique have been performed without a single major complication. Indirect decompression by LLIF provides a decompressive effect by stretching extensible structures such as the intervertebral disc and the flexible ligamentum flavum. Expanding the intervertebral height is also expected to have the effect of enlarging the intervertebral foramen. However, despite the belief that achieving an indirect decompression effect through ligamentotaxis requires the use of an appropriately sized cage, detailed reports on cage size and the indirect decompression have been limited [8,9]. Generally, the use of larger cages raises concerns about the risk of cage subsidence. The objective of this study is to identify the appropriate cage size and the extent of intervertebral height increase necessary to achieve sufficient indirect decompression. In this study, the ligamentotaxis effect was estimated by the measurement of the backward bulging length in MRI, pre- and post-operation. The relationships between cage size and backward bulging shortening, cage size and cage subsidence were examined.

## 2. Materials and Methods

### 2.1. Study Design

This is single-center, retrospective study. Materials are T2 images of 270 patients from June 2022 to March 2025, and 530 intervertebral spaces with spinal stenosis by herniated disc or spondylolisthesis. The inclusion criteria consisted of all cases and all intervertebral levels in which OLIF was performed for neurological symptoms caused by either central-type lumbar disc herniation or lumbar spondylolisthesis. Cases in which OLIF was performed solely for correction of lumbar alignment, such as scoliosis or kyphosis, without neurological symptoms due to anterior elements (e.g., intervertebral disc or posterior longitudinal ligament) compressing the dural sac, were excluded. The disc herniations included in this study were all extensive protruded hernias involving medial protrusion, and did not include paramedian small hernias, such as those treatable by simple herniotomy or endoscopy. Cases with only mild disc bulging and minimal posterior compression from discs were also not included in this study. Additionally, cases with sequestration herniation were not included in this study.

Data on gender, age, length of hospital stay, preoperative and postoperative lumbar JOA scores [10], and the level of the disease were collected. The present study and data collection were approved by the ethics committee of the Honjo Neurosurgery and Spine surgery clinic. StatMate 5 (Nihon 3B Scientific Inc., Niigata, Japan) was used for statistical analysis. Two-thirds of all surgical cases were performed by the author, while the remaining one-third was handled by another surgeon. All data collection and analysis was conducted solely by the author.

### 2.2. Measurements of Results

The measurement objects were the backward bulging length and the intervertebral height extension length. The definition for the backward bulging is the maximum posterior bulging length from the posterior surface of the vertebral body located forward at the level of the intervertebral disc (Figure 1).

If this backward bulging shortening was larger than 3 mm, we defined it as large shortening. We also measured the intervertebral height extension and divided it into these 3 groups: less than 3 mm, more than 3 mm, and 4 mm (Figure 2).

The occurrence of cage subsidence and its relationship with cage size were also investigated postoperatively. The follow-up period ranged from a minimum of 3 months to a maximum of 24 months. In this study, cases with a subsidence of 1/4 or more of the cage height were classified as having subsidence. The incidence of cage subsidence was examined between intervertebral spaces where a cage larger than 3 mm compared to the preoperative intervertebral height was used and those where it was not used.

### 2.3. Surgical Procedures

In order to minimize the impact on the iliopsoas and lumbar plexus, we perform oblique lateral interbody fusion (OLIF), prepsoaps approach. Set up in the right lateral decubitus position, muscle dissection and access were performed layer by layer under direct observation. Reaching the lateral aspect of the intervertebral disc by pulling the psoas muscle dorsally, a guide wire is inserted into the disc. After confirming through lateral fluoroscopy that the wire is positioned at near the center of the lateral aspect of the intervertebral disc, an access device is inserted to expand the surgical field. After clearing the disc space, trial cages are sequentially inserted, starting from 8 mm, to assess whether they are in the appropriate position and to confirm the fitting sensation. The cage size was determined based on the feel during trial cage insertion, ensuring that an appropriate size was selected without causing undue pressure. After inserting the appropriately sized cage, each muscle layer is sutured with fascia, and subcutaneous and skin closure is performed, concluding the lateral decubitus position procedure. Subsequently, a change in position to the prone position is carried out, and posterior fixation is performed using percutaneous pedicle screws.

## 3. Results

### 3.1. Patient Etiology

Out of 270 patients, 159 were male and 111 were female. The average age was 71.9 years (range 30–90), with a median age of 74 years. The youngest patient was 30 years old, and the oldest was 90 (Table 1). There were 201 cases of extensive protruded hernias involving medial protrusion and 121 cases of spondylolisthesis. In 52 cases, disc herniation was accompanied by spondylolisthesis at the affected intervertebral level. Most cases were affected by extensive protruded hernias or spondylolisthesis, leading to spinal canal stenosis.

Regarding the levels involved, there were two cases at L1/2, eighty-six cases at L2/3, two hundred cases at L3/4, and two hundred thirty-eight cases at L4/5. The average hospital stay for the cases was 9.6 days. The average preoperative lumbar JOA score was 14.4 (median 15), and it improved to an average of 23.6 (median 24) postoperatively.

### 3.2. Results of Backward Bulging Shortening and Cage Subsidence

Among all 530 intervertebral spaces, in 371 intervertebral spaces (70%), a cage 3 mm or larger than the preoperative intervertebral height could be inserted. In the remaining 159 intervertebral spaces (30%), cages of equal or less than 3 mm greater than the preoperative intervertebral height were inserted.

Among all 530 intervertebral spaces, significant shortening of the backward bulging length (3 mm or more) was observed in 353 intervertebral spaces. Among these, 339 intervertebral spaces were those where a cage 3 mm or larger than the preoperative intervertebral height was inserted. In only 14 intervertebral spaces, cages with a height less than 3 mm greater than the preoperative intervertebral height were inserted, but significant shortening of the backward bulging length was observed.

Only 8.8% of cases (14/159) with a large backward bulging shortening had an intervertebral height extension of 3 mm or less. On the other hand, a large reduction in backward bulging was observed in 91.4% of cases (339/371) with an intervertebral height extension of 3 mm or more, and in 92.0% of cases (229/249) with an intervertebral height extension of 4 mm or more.

These results suggest that the indirect decompression is likely to be obtained by a cage larger than 3 mm compared to the preoperative intervertebral height (*p* < 0.05). There was no statistically significant difference observed between cases using a cage larger by 3 mm or more compared to preoperative intervertebral height and cases using a cage larger by 4 mm or more.

In total, 17 patients (6.3%) required additional direct decompression in this series. In 14 cases, it was revealed that, in addition to the effects of an extensive protruded hernia or lumbar spondylolisthesis, there was significant bony lateral recess stenosis and ligamentum flavum hypertrophy. Therefore, direct decompression was performed simultaneously with LLIF. In three cases, symptoms did not sufficiently improve after LLIF through indirect decompression, so direct decompression was added. In all three cases, symptoms improved with direct decompression.

Postoperative cage subsidence was observed in 9.2% (49/530) of all intervertebral spaces, 8.6% (32/371) in spaces where a cage larger than 3 mm was used, and 10.7% (17/159) in spaces where smaller cages (not larger than 3 mm) were utilized. There was no statistically significant difference between these two groups. Among the cases that exhibited cage subsidence, there were no instances that required additional direct decompression after the initial surgery. In two cases, persistent low back pain was observed due to vertebral compression fractures caused by cage subsidence. Both patients had osteoporosis, with a young adult mean (YAM) of 70% or lower as measured by DEXA. Among the 107 cases treated after August 2024, 14 patients (13.1%) were diagnosed with osteoporosis based on DEXA. Of these fourteen patients, three patients (21.4%4) experienced cage subsidence.

### 3.3. Two Example of the Cases

Showcasing the measurement results of two representative cases, example case 1 (Figure 3) shows spinal stenosis with disc hernia and kyphosis L2-5. The backward bulging shortenings are all large (>3 mm) in these three intervertebral spaces (L2-5). Intervertebral height extension lengths are all quite large (>4 mm).

Example case 2 (Figure 4) reveals spondylolisthesis L4/5 with disc hernia L3/4. The backward bulging shortenings are large in L4/5 (>3 mm), but small in L3/4 (<3 mm). Intervertebral height extension lengths are more than 3 mm.

## 4. Discussion

These are multiple factors related to indirect decompression failure. Yngsakmongkol et al. reported several risk factors [2]. Low postoperative disc height is considered a risk. Except in cases of cage subsidence, the postoperative intervertebral height usually becomes the same as the height of the cage. The present results demonstrate that using a cage that is 3 mm or larger than the original intervertebral height yields a greater ligamentotaxis effect. By increasing the intervertebral height, soft tissues such as the intervertebral disc and the posterior longitudinal ligament are stretched vertically, resulting in a reduction in posterior compression. Since this ligamentotaxis effect also applies to central-type disc herniations, it can help decompress nerve compression caused by large disc herniations that protrude extensively into the spinal canal. In cases of lumbar spondylolisthesis, increasing the intervertebral height also reduces the anteroposterior displacement between vertebral bodies, thereby alleviating nerve compression. For foraminal stenosis, expansion of the intervertebral height directly enlarges the intervertebral foramen in the vertical direction.

In cases of spinal canal stenosis, if the cause is hypertrophied but soft ligamentum flavum or bulging of the intervertebral disc, symptom improvement can be expected through the indirect decompression effect achieved by restoring intervertebral height. However, when bony stenosis of the lateral recess is present or the ligamentum flavum has hardened due to calcification, the indirect decompression effect is less likely to be effective. Furthermore, in cases where the facet joints have undergone bony fusion, it is difficult to achieve sufficient intervertebral height extension, making the indirect decompression effect similarly unlikely.

Cage subsidence has been reported as a risk factor for indirect decompression failure. It is speculated that subsidence leads to a loss of vertebral height, weakening the ligamentotaxis effect and diminishing the enlargement of the intervertebral foramina. The occurrence of cage subsidence is believed to be influenced by various factors, including osteoporosis, intraoperative intervertebral manipulation, cage size, and other related factors. The assessment of bone quality, such as preoperative bone density, is considered important. While it is generally believed that using larger cages increases the likelihood of cage subsidence, this study did not observe a statistically significant difference. Among the series of surgical procedures from intervertebral disc curettage to cage insertion, the most likely to cause endplate injury is the technique of penetrating the disc tissue from the left to the right side near the vertebral endplate using instruments such as a Cobb spinal elevator in combination with a hammer or the technique of forcefully inserting a trial cage into the intervertebral space. Most endplate injuries that lead to cage subsidence are thought to occur during these steps. Therefore, these procedures should be performed carefully, with fluoroscopic confirmation of the instrument’s position and angle. Additionally, during the insertion of the trial cage or LLIF cage, the intervertebral space must be adequately cleared of disc material. If disc tissue remains unevenly, the cage may not be inserted at an appropriate angle, increasing the risk of vertebral endplate injury and subsequent cage subsidence. Attention must also be paid to facet joint bony fusion. When posterior bony fusion is present, even if a larger cage is inserted with the expectation of achieving indirect decompression, the intervertebral height may not expand due to the fusion, potentially resulting in cage subsidence. Therefore, it is important to evaluate preoperative CT scans, particularly sagittal views, to confirm the presence or absence of facet joint fusion. In the present series, cage subsidence occurred in 9.2% of all treated intervertebral levels; however, no cases required additional direct decompression due to a diminished indirect decompression effect. Nevertheless, in cases where cage subsidence led to vertebral body deformation, prolonged low back pain was observed, indicating the need for caution regarding cage subsidence.

As a result, in our case series, there were only three cases that required additional direct decompression and eleven cases that required direct decompression simultaneously with LLIF. The majority of cases, other than the 14 instances that required direct decompression, have shown symptom improvement and a favorable postoperative course without the need for direct decompression. Indirect decompression is thought to work by reducing anterior–posterior protrusion by stretching soft tissues such as the intervertebral disc, posterior longitudinal ligament, and ligamentum flavum vertically. Therefore, in cases where there is primarily bony spinal canal stenosis, the effects of indirect decompression are not realized. Furthermore, in cases of significant ligamentum flavum hypertrophy or lateral recess stenosis, even if indirect decompression occurs, it may not provide sufficient decompressive effects on the nerve roots. Instead, the elongation of the intervertebral height could stretch the nerve roots, potentially worsening symptoms such as neuralgia. Therefore, in cases of bony spinal canal stenosis, lateral recess stenosis, or significant ligamentum flavum hypertrophy, it is considered necessary to combine direct decompression with the procedure.

The results of this study indicate that using a cage with a height 3 mm or more greater than the original intervertebral height is effective in achieving a favorable ligamentotaxis effect on imaging. Based on these findings, selecting a cage size that allows an increase of 3 mm or more in intervertebral height is considered effective. The decision to use a cage that is 3 mm or more larger is made based on factors such as the sensation during the insertion of the trial cage into the intervertebral space, bone density, facet joint bony fusion, and overall safety considerations. It is also important to evaluate preoperatively using CT images whether the cage can be inserted safely and without undue force, based on the insertion angle and its spatial relationship with the iliac crest. We use a larger cage only when it is deemed fitting and can be done safely in a comprehensive manner.

Moreover, the primary purpose of using a larger cage is to expand the intervertebral height. Therefore, in patients whose vertebral endplates are not flat but have a concave, curved shape centrally, it is not necessarily required to add an additional 3 mm to the preoperatively measured intervertebral height. In cases where the endplate contour is curved and degenerative changes have caused the rims of the upper and lower vertebral bodies across the disc space to be in close contact, sufficient intervertebral height expansion can be achieved with a minimal-height cage—as long as the inserted cage adequately supports both lateral rims of the vertebral bodies. Indeed, the fact that a significant ligamentotaxis effect was observed in 8.8% of cases even with cages that were not substantially larger than the preoperative intervertebral height (i.e., +3 mm or less) is likely attributable to the reasons mentioned above. In other words, to safely achieve an intervertebral height expansion of more than 3 mm, it is essential to thoroughly assess factors such as the vertebral bone morphology on preoperative CT images, the degree of endplate sclerosis, and the extent of residual intervertebral disc tissue.

The limitation of this study is that the follow-up period varied among cases, ranging from 3 months to 1 year. Therefore, the results regarding the risk of cage subsidence may change as the follow-up period is extended in the future.

## 5. Conclusions

To achieve a sufficient ligamentotaxis effect, it is necessary to select a cage size that allows for an intervertebral height increase of at least 3 mm compared to the preoperative measurement. The risk of cage subsidence associated with the use of larger cages will require further investigation through longer-term follow-up.

## Figures and Tables

**Figure 1 jcm-14-04554-f001:**
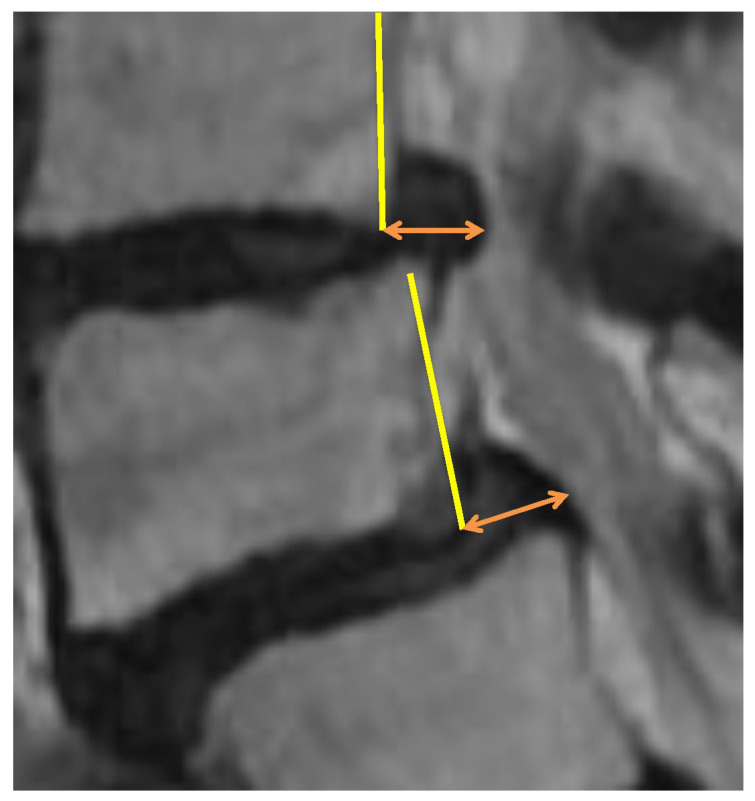
Definition of the “backward bulging” is posterior bulging length (orange arrow) from the posterior surface of the vertebral body (yellow line) at the level of the intervertebral disc.

**Figure 2 jcm-14-04554-f002:**
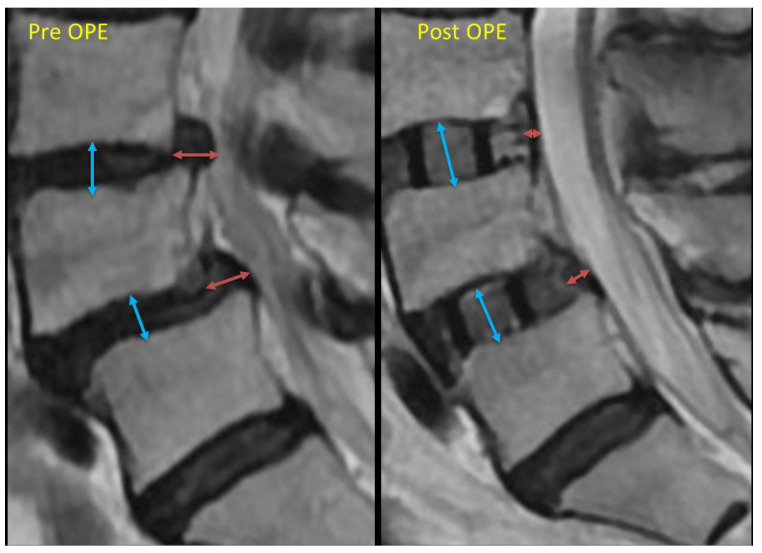
This is the schema of the “backward bulging” length measurement. The red arrow indicates the backward bulging length. We measured the difference between pre- and post-operation measurements. If this backward bulging shortening exceeded 3 mm, we defined it as significant shortening. The blue arrow indicates the “intervertebral height”. Post-operation, this height equals the size of the cage. We also measured the difference as the “intervertebral height extension”. These measurements were then divided into three groups: less than 3 mm, more than 3 mm, and 4 mm.

**Figure 3 jcm-14-04554-f003:**
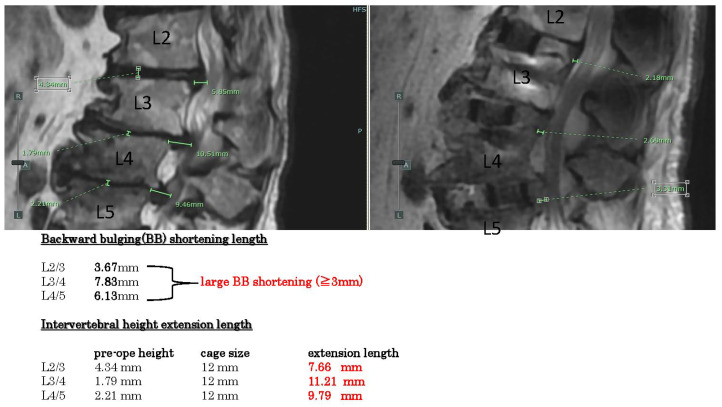
Example case 1, spinal stenosis with disc hernia and kyphosis. Backward bulging shortenings are all large in these 3 intervertebral spaces. Intervertebral height extension lengths are all large.

**Figure 4 jcm-14-04554-f004:**
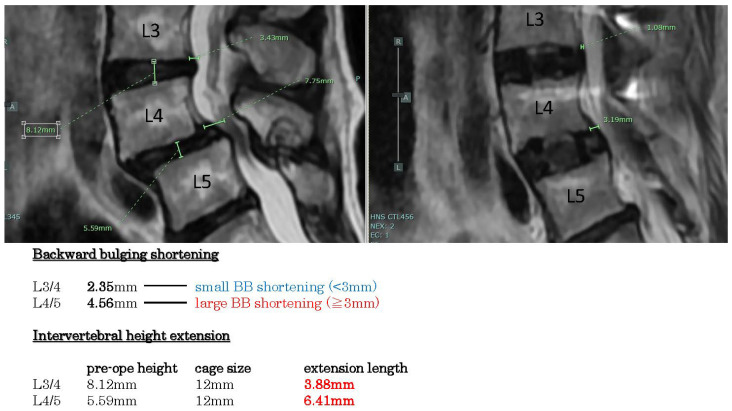
Example case 3, spondylolisthesis with disc hernia. Backward bulging shortenings are large in L 4/5, but small in L3/4. Intervertebral height extension lengths are more than 3 mm.

**Table 1 jcm-14-04554-t001:** Patients etiology and Results.

All cases: n	270
Male: female	159:111
Age: average (median, range)	72.0 (74, 30–90)
Cases with extensive protruded hernia: n	201
Cases with spondylolisthesis: n	121
L1/2 level: n	2
L2/3 level: n	86
L3/4 level: n	200
L4/5 level: n	238
Hospital stay days: average (median, range)	9.6 (9, 4–28)
Preoperative lumbar JOA score: average (median, range)	14.4 (15, 0–27)
Postoperative lumbar JOA score: average (median, range)	23.6 (24, 8–29)
All intervertebral spaces: n	530
≥3 mm larger cage inserted: % (n)	70% (371/530)
<3 mm larger cage inserted: % (n)	30% (159/530)
Intervertebral spaces with large shortening of the BB length(≧3 mm): n	353
≥3 mm larger cage inserted: % (n)	96.0% (339/353)
<3 mm larger cage inserted: % (n)	4.0% (14/353)
Large BB shortening rate: % (n)	
<3 mm intervertebral height extension	8.8% (14/159)
≥3 mm intervertebral height extension	91.3% (339/371)
≥4 mm intervertebral height extension	92.0% (229/249)
Cases which required additional direct decompression: % (n)	6.3% (17/270)
Cage subsidence rate of all intervertebral spaces: % (n)	9.2% (49/530)
Cage subsidence rate where ≥3 mm larger cage was used: % (n)	8.6% (32/371)
Cage subsidence rate where <3 mm larger cage was used: % (n)	10.7% (17/159)

BB: backward bulging.

## Data Availability

Data in this article is available upon request to interested researchers.

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
