# Peer review of "Ligamentotaxis Effect of Lateral Lumber Interbody Fusion and Cage Subsidence"

_jcm, 2025, doi:10.3390/jcm14134554_

Round 1
Reviewer 1 Report
Comments and Suggestions for Authors
Dear Author,
The article is interesting.
Here are my points:
- Title – no point is needed after the title.
- Methods – this is a single center experience and it should be mentioned from the start in addition to the retrospective design.
- Methods – the ethical approval of retrospective data collection is needed since the procedures have been done by different surgeons.
- Results – some information with respect to the study population is needed (e.g. age – mean or ranges, etc.)
- Results. – the last statement is not clear with concern to the two subgroups
- Conclusion – are there any considerations regarding the long-term post-surgery outcome and further research to confirm it?
- Keywords – please remove “keyword 1”
- Introduction – please add some data with respect to the safety of the mentioned procedures and current guideline level (if any)
- Introduction – pros and con, pitfalls or challenges regarding the procedure should be briefly mentioned from the start.
- Objective – this should be a distinct section at the end of Introduction before specifying the exact measurements (e.g. to analyze….or to evaluate..)
- Methods - the first subsection is the study design (e.g. retrospective, single-center study), followed by the study population (inclusion and exclusion criteria), followed by the data collection/assessments/measurements, followed by the statistical approach, followed by the ethical aspects. These subsections should be very clear and distinct.
- Typo – a space is needed at “Figure1”
- “Figure 2 is schema….” = The text should be clearly specified and the figure helps the understanding of the text, not the text helps the understanding of one figure.
- Page 4, line 101. “We performed…” is not adequate since this seems (until this point of the article) this is a retrospective analysis. Moreover, if “we” is adequate, why a single author?
- The text body just be Justify
- Table 1 should accompany the data presentation in the Results section
- “representative cases” – this not clear. Did you present the entire study population and selected a few cases? In this instance, we should mention at Methods why these cases have been selected.
- Results section should be re-organized. Please present the entire study results, and then select the particular aspects.
- Discussion section introduces data from literature versus these original results. This means that a larger body of evidence is needed when compare your data.
- The last subsection at Discussion is the limitations of the current study.
- Conclusion requires a few clear statements as a take-home message based on your original data and maybe further expansion
- The country and the city are not specified at affiliation, neither at Institutional Review Board Statement.
- It is not clear the role of the author – the author is the single surgeon in this analysis or just collected the data? Is this a single-center experience of several physicians?
- As I mentioned, the number of references should be expanded.
Thank you
Author Response
Revisions were made as each indicated points.
please check attached Word file.
Responses to each comment number are provided below.
1. Revisions were made as indicated.
2.Revisions were made as indicated.
3.Revisions were made as indicated.Two-thirds of all surgical cases were performed by the author, while the remaining one-third were handled by another surgeon. All data collection and analysis were conducted solely by the author.
4.Revisions were made as indicated.
5.Additional content was included to clarify the intended meaning in last sentence of Results.
6.Revisions were made as indicated.
7.Revisions were made as indicated.
8.Revisions were made as indicated.
9.Revisions were made as indicated.
10.Revisions were made as indicated.
11.Revisions were made as indicated.
12.Revisions were made as indicated.
13.Revisions were made as indicated.
14.Revisions were made as indicated.
15. This text body style is MDPI’s template.
16. The content listed in Table 1 is explained sequentially in the Results section.
17.The two cases were included as practical examples of the measurements conducted in this study. They are not representative cases per se, but rather intended to help readers better understand the measurement methods. These two cases have been moved to the end of the Results section.
18.As suggested, the Results section has been reorganized so that the overall findings are presented first.
19.I apologize, but I am unable to find any previous studies that directly compared cage size with the effect of indirect decompression. Therefore, a direct comparison with the present study is difficult.
20. Revisions were made as indicated.
21. Revisions were made as indicated.
22.Revisions were made as indicated.
23.Two-thirds of all surgical cases were performed by the author, while the remaining one-third were handled by another surgeon. All data collection and analysis were conducted solely by the author. Revisions were made as indicated.
24.The reference list has already been expanded to include 10 sources, and I do not believe that further additions are necessary.

Reviewer 2 Report
Comments and Suggestions for Authors
Ligamentotaxis effect of lateral lumber interbody fusion and cage subsidence.
Outline: This study aimed to assess the ligamentotaxis effect by measuring the "back-ward bulging" length in pre and postoperative MRIs and examining its correlation with cage size and subsidence. From this study, the authors concluded that selecting a cage larger than 3mm compared to the intervertebral height is likely to achieve an effective ligamentotaxis effect. Surgical deci-sions should consider preoperative images and safety considerations for optimal out-comes.
Critique:
1. The format needs to be fixed.
- Introduction:
. Please re-structure for this study. From basic concept to specific aim.. It needs to be specified..
- Method:
. Before measurement of results, patients recruitment including inclusion/exclusion criteria is essential for this study.
. Furthermore, please define the measurement. Please add basical parameters such as lumbar lordosis and spinopelvic parameters
. The authors did not describe the statistical analysis
. Surgical procedurs: How did the authors determined the cage size?
- Results:
3.2 section should not be comprised as main structure.
For the results, the authors need to be re-structure. For example, not descirbe "Results of 530 intervertebral spaces of 270 cases." It need to be related with subject, such as "backward bulging shortening"
- Discussion:
The authors did not well described the results, and comparison with previous results.
Author Response
Comment1 : Introduction Please re-structure for this study. From basic concept to specific aim.. It needs to be specified..
Response: Revisions of the introduction part was made as indicated.
Comment2:
. Before measurement of results, patients recruitment including inclusion/exclusion criteria is essential for this study.
Response2:
Inclusion/exclusion criteria is added in 1st paragraph of Results.
Comment3: Furthermore, please define the measurement. Please add basical parameters such as lumbar lordosis and spinopelvic parameters
Response3: Parameters related to lumbar lordosis or spinal alignment were not addressed in this study. The outcome measures of this study are limited to the evaluation of indirect decompression effects at the treated local intervertebral levels. This is because including alignment analysis would broaden the scope excessively and detract from the study’s primary focus. Evaluation of alignment parameters will be conducted in a separate future study.
Comment 4. The authors did not describe the statistical analysis
Response4. Statistical analysis was described in Results.
“These results suggest that the indirect decompression is likely to be obtained by a cage larger than 3 mm compared to the preoperative intervertebral height (p<0.05). There was no statistically significant difference observed between cases using a cage larger by 3mm or more compared to preoperative intervertebral height and cases using a cage larger by 4mm or more.”
Comment 5
Surgical procedurs: How did the authors determined the cage size?
Response 5
Ultimately, the decision is based on the tactile feedback obtained during the insertion of the trial cage.
Comment 6
Results:
3.2 section should not be comprised as main structure.
For the results, the authors need to be re-structure. For example, not descirbe "Results of 530 intervertebral spaces of 270 cases." It need to be related with subject, such as "backward bulging shortening"
Response 6
Revisions were made as indicated.
Comments7.
- Discussion:
The authors did not well described the results, and comparison with previous results.
Response 7
Revisions were made as indicated.The Discussion section has been substantially expanded, so we would appreciate it if you could review it. As for the comparison with previous studies, I apologize, but I anm unable to find any previous studies that directly compared cage size with the effect of indirect decompression. Therefore, a direct comparison with the present study is difficult.

Round 2
Reviewer 1 Report
Comments and Suggestions for Authors
Dear Author,
Well done!
Reviewer 2 Report
Comments and Suggestions for Authors
Manuscript substantially revised and I have no more comment